# The Association between Personality Factors and Metabolic Parameters among Patients with Non-Alcoholic-Fatty Liver Disease and Type 2 Diabetes Mellitus—A Cross-Sectional Study

**DOI:** 10.3390/jcm12134468

**Published:** 2023-07-03

**Authors:** Marcin Kosmalski, Rafał Frankowski, Monika Różycka-Kosmalska, Kasper Sipowicz, Tadeusz Pietras, Łukasz Mokros

**Affiliations:** 1Department of Clinical Pharmacology, Medical University of Lodz, 90-153 Lodz, Poland; tadeusz.pietras@umed.lodz.pl; 2Students’ Research Club, Department of Clinical Pharmacology, Medical University of Lodz, 90-153 Lodz, Poland; rafal.frankowski@stud.umed.lodz.pl; 3Department of Electrocardiology, Medical University of Lodz, 92-213 Lodz, Poland; monika-rozycka-kosmalska@umed.lodz.pl; 4Department of Interdisciplinary Disability Studies, The Maria Grzegorzewska University in Warsaw, 02-353 Warsaw, Poland; ksipowicz@aps.edu.pl; 5Second Department of Psychiatry, Institute of Psychiatry and Neurology, Sobieskiego 9, 02-957 Warsaw, Poland; lmokros@ipin.edu.pl

**Keywords:** diabetes mellitus type 2, non-alcoholic fatty liver disease, personality factors, emotional intelligence, coping style

## Abstract

Background: The increasing prevalence of type 2 diabetes mellitus (T2DM) and non-alcoholic fatty liver disease (NAFLD) states a serious problem for public health. The introduction of effective methods of treatment and prevention is crucial to avoid complications of these diseases. Among them, we can specify psychological factors that affect everyday life and determine the patient’s attitude towards therapy, and what follows, their compliance in treatment. The literature indicates these connections in various ways; in our study, we extend this view to include a broader perspective of human personality. Objective: We decided to investigate the associations between personality factors and metabolic parameters in patients with NAFLD and T2DM in order to better understand the patient’s approach to the treatment of a chronic disease, such as those mentioned, and to establish the basis for further research implementing psychological interventions in the treatment of NAFLD and T2DM. Methods: One hundred participants with NAFLD and T2DM underwent blood tests and anthropometric measures. Each of them was asked to complete five questionnaires evaluating their personality properties. Results: We revealed that a rise in body mass index is related to a fall in the emotional intelligence factor of utilizing emotions, and a rise in emotional perception. The decrease in task-oriented coping style and a rise in emotion-oriented coping style are associated with a waist-hip ratio increase. The increase in fasting plasma glucose is predicted by a decrease in task-oriented coping style score. A fall in social diversion coping style score is associated with a high-density lipoprotein increase; in turn, a triglycerides increase is connected with a decline in rhythmicity score. Conclusions: The personality factors are in relationship in the management of NAFLD and T2DM. They affect a patient’s approach to treatment, which is very important, because we know lifestyle and dietary interventions are an important part of the treatment of these diseases. The compliance manifests by lifestyle modifications, taking medications regularly, measuring blood glucose, and inspection visits in outpatients’ clinics are a large part of a diabetic’s life. Future studies introducing psychological intervention to improve, e.g., coping styles or rhythmicity are needed to implement new methods of patient management.

## 1. Introduction

Diabetes mellitus (DM) is a group of metabolic disorders that are a result of environmental and hereditary factors contributing to the progressive loss of pancreatic beta cells and impairment of their functions. It is considered that DM is a serious threat and problem for public health worldwide, with a prevalence of about 10.5% and rising. Type 2 diabetes (T2DM) is responsible for 90–95% of all cases. T2DM, in its course, entails lots of complications [1,2]. As T2DM is a metabolic disease, it may be related to other metabolic disorders, such as obesity or non-alcoholic fatty liver disease (NAFLD). Among subjects with T2DM, the prevalence of NAFLD is increased and reaches about 55% vs. circa 32% in the general population. On the other hand, among subjects with NAFLD, T2DM is more frequent than in non-NAFLD subjects and reaches about 45% [3,4,5]. What is important to say is that T2DM may be a significant risk factor of liver fibrosis development [6].

NAFLD is the most common chronic liver disease. It is diagnosed when the count of injured hepatocytes affected by steatosis reaches at least 5% without detection of any secondary cause of this state. At its course, it brings serious complications, including its progression to non-alcoholic steatohepatitis, cirrhosis, and even hepatocellular carcinoma, which is the most frequent tumor among these disease consequences, but also other malignant tumors or cardiovascular diseases [7]. The NAFLD treatment is limited to lifestyle modifications, including diet and physical activity [8]. Stewart et al.’s [9] study confirmed that personality factors also play a role in dealing with NAFLD [9]. NAFLD may be in relationship with mental disorders, including depression, which may be related with nutritional patterns, lifestyle, and environmental traits [10,11]. Personality seems to be a connector between NAFLD and psychiatric disorders due to its impact on lifestyle, weight changes, scrupulousness, and even immune system function [11]. Anxiety and depression are considered risk factors for NAFLD. Also, cognitive and behavioral factors, including the ability to change and physical activity are the issues which may alleviate both NAFLD and depression [12]. Personality can be defined as a complex set of factors that greatly influences a person’s life and behavior [13]. In a human lifespan, almost all aspects of human functioning are influenced by personality, which comprises a pattern of mainly unconscious and relatively stable, deeply ingrained psychological characteristics. Thus, personality could be described as a working and associating style pattern and set of subject features that are defined by staying in relationships, emotions, habits, and attitudes [14]. Also, human goals, dispositional traits, and integrative life stories are part of it. The development of personality is affected by multiple factors throughout life, such as social functions and environmental influences. It was revealed that human temporal stability is directly proportional to age [15]. It is known that psychological aspects affect subject behavior and bring about biophysical changes [16]. Proper nutrition, physical activity, glycemic level measurements, and taking medicines are necessary to maintain adequate glycemic control, and improve patients’ well-being [17]. A positive correlation between psychological factors and diabetes control or NAFLD management has been demonstrated, which is in line with treatment compliance [9,18]. From the clinical point of view, obtaining rewarding compliance in such diseases treatment is very important due to the nature of these disorders and the need to involve the patient in therapy so that he maintains his diet and physical activity [19]. It was revealed that alterations in emotional distress and behavioral skills and coping influence the medication adherence [20].

Coping mechanisms have an impact on the patient’s behavior in specific situations, including his approach to the disease. Generally, coping refers to avoiding or managing distress [21]. We can divide coping strategies into some categories in multiple ways. The division of coping skills into problem-focused, emotion-focused, and avoidance skills is popular. The first refers to alleviating the effects of stress through facing and solving it; the second could be described as the regulation of emotional distress; and the last one relies on denying, minimizing, and avoiding stressors [17,22]. There are also other classifications of coping strategies, divided into active, which refers to facing the stressors, and passive, which refers to avoiding them or reducing their effects through some behaviors [23].

Diabetes affects multiple lifestyle areas, and results in an increase of perceived stress and a fall in self-acceptance. Active coping abilities may have a beneficial effect on self-esteem; in turn, maladaptive coping could exacerbate everyday stress, which is related to diabetes mellitus [17].

Temperament is characterized as the biologically-determined, relatively constant part of the personality. It was revealed that certain features measured by the Temperament and Character Inventory (TCI), developed by Cloninger, affect compliance to drug follow-up [24]. Yoda et al. [18] divided the patients with T2DM into two clusters based on TCI. They revealed that resistance to change and a lack of cooperativeness correlated positively with hemoglobin A1c (Hb1Ac) levels, which is a marker of glycemic control [18]. What is known, is that compliance decreases with the patient’s age [25]. It was proven that HbA1c level is linked to emotion regulation and, as a consequence, with emotional intelligence (EI) [26]. The importance of patient awareness and cooperation is underscored by the fact that self-care education positively affects EI and HbA1c levels [27]. The literature mainly refers to the influence of temperament [25,28] and stress coping style [29,30,31] in diabetes mellitus control, which is associated with compliance in treatment. As personality is a broader concept, it is interesting whether various temperament conceptual frameworks presented, e.g., by Eysenck or Strelau are related to metabolic parameters in patients with T2DM. However, work in this area is lacking.

Therefore, the purpose of this study was to explore the relationship between personality factors and chosen metabolic parameters among patients diagnosed with T2DM and NAFLD, independently of the current hypoglycemic pharmacotherapy. To our knowledge, there are no studies on the personality factors within NAFLD and T2DM-diagnosed patients, and our work is the first to address this topic.

## 2. Materials and Methods

### 2.1. Ethical Approval

This study was conducted in accordance with the Declaration of Helsinki. The study was approved by the local Ethics Committee (Consent of Research Review Board of the Medical University of Lodz, Lodz, Poland; no. RNN/181/17/KE).

### 2.2. Participants

One hundred patients diagnosed with T2DM and NAFLD were recruited between 2019 and 2021 from patients of the Specialist Outpatient Clinic of Diabetes at Norbert Barlicki Memorial Teaching Hospital No. 1 of The Medical University of Lodz. The group comprised sixty-five females (mean age = 65.39 ± 8.82) and thirty-five males (mean age = 61.77 ± 7.94). All participants gave their written informed consent to take part in the study at the time of registration. All the participants were recruited voluntarily and had an opportunity of withdrawing the informed consent at any point of the study. Inclusion criteria were: confirmed diagnoses of both T2DM and NAFLD. Exclusion criteria included: secondary reasons of increased fatty liver accumulation, such as the use of combined antiretroviral treatment and ethanol use over 14 g each day, other medications use including amiodarone, chemotherapy, tamoxifen, methotrexate, corticosteroids, tetracyclines, amphetamines, acetylsalicylic acid, valproic acids; gastroenterological reasons: malnutrition, starvation, total parenteral nutrition, celiac disease, severe surgical weight loss, pancreatectomy, short bowel syndrome; other reasons: hemochromatosis, Wilson’s disease, a lack of alpha-1 antitrypsin, glycogen storage related diseases, lysosomal acid lipase deficiency, familial hyperlipidemia, hereditary fructose intolerance, abundances in urea cycle, lack of citrin, hypobetalipoproteinemia, abetalipoproteinemia, congenital lipodystrophy, hypothyroidism, pituitary and hypothalamic malfunction, polycystic ovary syndrome, HELLP (hemolysis, elevated liver enzymes and low platelets), growth hormone deficiency, environmental factors: cadmium, lead, arsenic, mercury, herbicides, pesticides, polychlorinated biphenyls, chloroalkenes, phosphorus or Amanita phalloides poisoning, petrochemicals and Bacillus cereus, other than T2DM, prediabetes, severe infections, kidney or liver dysfunction (including those who had an organ transplant), recent surgery, musculoskeletal damage within the last six months, pregnancy, and diseases that may affect muscle metabolism, such as lipid metabolism disorders, mitochondrial myopathies, and disorders of glycogen metabolism.

### 2.3. Measures

The study used the results of physical examination, laboratory and anthropometric tests ordered during standard follow-up visits in patients with T2DM and NAFLD diagnosis. The ADA-recommended diagnostic procedures for anomalies in glucose metabolism or the existence of secondary causes of fatty livers were carried out. Performed laboratory and anthropometric tests included measurement of systolic and diastolic blood pressure (SBP and DBP, respectively), height, waist circumference (WC), hip circumference (HC), and body weight. Waist-hip ratio (WHR) and body mass index (BMI) were calculated from outcomes of anthropometric measures. Blood cell count (to assess blood platelets—PLT), glycated hemoglobin (HbA1c), fasting plasma glucose (FPG), uric acid, creatinine, total cholesterol (TC), low density lipoprotein cholesterol (LDL), high density lipoprotein cholesterol (HDL), triglyceride (TG) concentrations, alanine aminotransferase (ALT), asparagine aminotransferase (AST), and gamma-glutamyltransferase (GGTP) activity were all measured via blood tests. Standard laboratory procedures were used to measure these markers. Two hours after a standardized meal, or around 20% of their total daily calorie needs, postprandial plasma glucose (PPG) was assessed. Standard laboratory procedures were used to measure these markers. A skilled radiologist also conducted an ultrasound examination (US) to determine whether hepatic steatosis was present. The presence of localized fat sparing, decreased conspicuity of the hepatic vasculature, increased liver echogenicity in comparison to the renal cortex, and decreased visual acuity of the deeper liver parenchyma were all considered indicators of hepatic steatosis. To confirm the presence of fatty liver Hepatic Steatosis Index (HSI) and Fatty Liver Index (FLI) were calculated. To help determine NAFLD/NASH status (presence of liver fibrosis), we used Fiborisis-4 Index (FIB-4).

To determine HSI, the following formula was used:

8 × ALT/AST + BMI (+2 if T2DM yes, +2 if female). BMI was calculated using: body weight (kg)/height squared (m2). HIS values at least 36 or above is a sign of a high probability of NAFLD [32].

To determine FLI, the following formula was used:(e0.953 × loge (TG) + 0.139 × BMI + 0.718 × loge (GGTP) + 0.053 × WC − 15.745)/(1 + e0.953 × loge (TG) + 0.139 × BMI + 0.718 × loge (GGTP) + 0.053 × WC − 15.745) × 100.

FLI ≥ 60 indicate that NAFLD is present [33].

To determine FIB-4, the following formula was used:Age [years] × AST [U/l]/(platelets [×10^9^/L] × √(ALT [U/l])).

FIB-4 < 1.3 indicate a low risk of liver fibrosis [34,35].

Patients who lacked fatty liver evidence in all of these studies’ (HSI, FLI, and US) were not enrolled, due to questionable NAFLD diagnosis.

Eysenck Personality Questionnaire—Revised (EPQ-R) in Polish adaptation by Jaworowska was utilized to measure personality according to Eysenck. It consists of scales representing the three PEN superfactors (Psychoticism, Extraversion, and Neuroticism) and the Lie scale [36]. The Impulsiveness Questionnaire (IVE) was used to assess three additional personality traits according to Eysenck, namely Impulsiveness, Venturesomeness, and Empathy [37]. Emotional Intelligence was measured with the Emotional Intelligence Questionnaire (INTE) by Jaworowska and Matczak [38]. Its score is either a general measure or two subscores: utilization and perception of emotions. Stress coping was assessed with the Endler’s and Parker’s Coping Inventory For Stressful Situations (CISS) in Polish normalization by Strelau and Jaworoska. The questionnaire comprises three scales representing key coping styles: Task-oriented, Emotion-oriented, and Avoidance, with the latter being divided into Distraction and Social diversion [39]. The Functional Characteristics of Behavior–Temperament Inventory Revised (FCZ-KT RI) was created and validated by Cyniak-Cieciura, Zawadzki, and Strelau. It has seven scales, with four signifying the energetic behavioral characteristics (Sensory Sensitivity, Endurance, Emotional Reactivity, Activity), and three illustrating the temporal features of behavior (Briskness, Perseverance, Rhythmicity) [40].

The results of the laboratory, anthropometric measures, and questionnaires were analyzed by researchers with adequate competence.

### 2.4. Statistical Analysis

The statistical analysis was conducted in Jasp 0.16.4 (University of Amsterdam, The Netherlands). Each binominal variable was presented as prevalence in the sample, as both absolute (number of observations, N) and relative scores (percentage, %). Each continuous variable was characterized as mean with standard deviation (SD). For continuous variables, the distribution was assessed based on the Shapiro–Wilk W-test, the Q-Q plots of the residuals. The heterogeneity of variance between the subgroups was confirmed with Levene’s test. Intergroup comparisons were conducted by analysis of variance: the Snedecor’s F test or Welch’s test, depending on the result of Levene’s test. The Chi-square test was performed to assess contingencies in 2 × 2 tables (comparison of categorical variables of interest between faculties). If the expected value in any of the cells of the 2 × 2 tables was below five, two-tailed Fisher’s exact test was used instead of the Chi-square test. 

Multivariate linear regression models were constructed to further assess the predictive value of personality factors. A separate model was constructed for each of the selected metabolic parameters. Each model was adjusted for sex, duration of the diabetes, and current hypoglycemic pharmacotherapy. A global Fisher–Snedecor test was used to assess the goodness of fit of the model. An analysis of residuals was performed to assess the validity of assumptions of normality, homoscedasticity, and independence between observations (with the Durbin–Watson test). Tolerance indices were analyzed to track possible multi-collinearities—a lack of significant collinearity was adopted for a tolerance index > 0.1.

A residuals analysis was conducted to evaluate the validity of the assumptions of normality and independence between observations (with the Durbin–Watson test). The tolerance indices were examined in order to identify potential multi-collinearities. Effect sizes were assessed in two manners: for a model as a whole (coefficient of determination R^2^) and for each parameter in the model (semi-partial correlation sR). The level of significance was adopted for α = 0.05.

### 2.5. Data Availability Statement

The database used to support this study’s findings may be obtained upon request to the corresponding author.

## 3. Results

### 3.1. Study Group Characteristics and Intergroup Comparison

The study group comprised 65 women and 35 men. The mean age was higher for women than for men. Regarding the assessed clinical parameters, men had a higher waist-hip ratio and diastolic blood pressure. No other differences between groups were statistically significant (Table 1). In terms of personality factors, women scored higher than men on the EPQ-R Lie scale, FCZ-KT emotional reactivity, IVE Empathy scale and lower than men on IVE Venturesomeness scale. The differences regarding the remaining assessed personality features were statistically insignificant (Table 2).

### 3.2. Multivariate Linear Regression Models—Prediction of the Metabolic Parameters with Personality Factors

A rise in BMI was related to a fall in the emotional intelligence factor of utilizing emotion and a rise in emotional perception.

An increase in WHR was associated with a decrease in task-oriented and avoidance coping styles, and a rise in emotion-oriented coping style.

A rise in fasting glucose was predicted by a decrease in task-oriented stress coping style score.

An increase in HDL concentration was associated with a fall in social diversion coping style score.

An increase in TG was related to a fall in Rhythmicity score.

An increase in the FIB-4 score was linked to a fall in endurance and social diversion.

No other association between personality factors and selected metabolic parameters was found statistically significant (Table 3).

## 4. Discussion

The main hypothesis of our study was that personality factors affect metabolic parameters in NAFLD and T2DM-diagnosed patients. In light of the results of our study, the importance of the impact of psychological factors on the metabolic parameters of patients with NAFLD and T2DM should be emphasized. We found that task-oriented (or problem-solving) coping styles were associated with better WHR and fasting glucose, while emotion-oriented coping styles were associated with an increase in WHR. The WHR increase was also related to a decrease in the avoidance coping score. This is in line with previous findings, since the configuration of high problem-focused and low emotion-focused coping tendencies is associated with active coping and improved mental and physical health outcomes [17,23,30,41]. Problem solving coping style was also previously associated with better well-being in the course of T2DM [21].

Our study revealed connections between psychological factors and lipid profiles. A decrease in the social diversion coping style score was linked to an increase in HDL levels. Ariaratnam et al. [42] have previously found a link between coping styles and the lipid profile. This study revealed the beneficial effect of an avoidance-oriented coping style on TC and LDL in a healthy subject group [42]. We also found a relationship between rhythmicity and the level of triglycerides. The possible explanation of this result is that rhythmicity may exert itself in a tendency toward regular hours of meals and physical activity. As we know, managing diabetes requires controlling the glycemia levels, which relates to taking regular medicines and eating at the right times, not snacking at night. This result corresponds with Park et al.’s [43] study, which revealed that following the regular diet has a beneficial impact on body weight, better nutrition status, WC, BMI, blood pressure, and glycemic and lipid management parameters [43]. What is in line with rhythmicity and diabetes mellitus treatment is regular physical activity, which has beneficial effects on lipid metabolism, insulin sensitivity, plasma glucose level, and waist circumference [44,45].

The relationship between BMI and emotional status is well known [46]. It was proven that overweight subjects have higher emotional attention and lower emotional repair abilities [47]. We found a raise in the BMI parameter in correlation with a decrease in the emotional intelligence factor of utilizing emotion and a rise in emotional perception. As the emotion utilizing is higher, the BMI is lower, which could be explained by higher motivation to follow up on a diet or not react by overeating or stress-eating. In turn, an increase in perception of emotion (thus, its feeling) was linked to a rise in BMI, independently of emotion utilizing. This phenomenon could be related to high emotional distress, followed by emotion suppression (because emotion is not effectively processed) and overeating or stress-eating. A similar pattern has been seen among overweight individuals and adolescents [48,49]. However, the data on emotional eating and its influence on BMI is contradictory. Some authors prove that emotional eating results in BMI elevation, while others do not prove this thesis [50,51].

Previously, it was shown that emotional intelligence and coping styles affect glycemic control, as illustrated by HbA1c levels [26,52]. However, our study did not confirm these connections.

An undoubted advantage of the work is a comprehensive assessment of the patients’ personality, which is an attempt to take a broad look at the personality factors determining the course of T2DM. What follows this personality assessment and psychological intervention seems to be a hopeful strategy for increasing the patient’s compliance to follow up on the treatment, or to assess expected cooperation at the beginning of treatment.

## 5. Limitations

Several restrictions should be mentioned regarding the study’s methodology and the interpretation of its results. The sample was relatively small, non-random, and originated from a single center. This hampers the generalizability of the results. However, the sample represented a naturalistic, clinical group. Also, the patients were given strict considerations regarding the exclusion criteria. The huge number of statistical tests conducted constituted a risk of type I error, but the exploratory nature of the study necessitated abandoning certain adjustments. Another limitation is the purely observational character of the study. A prospective, multicenter trial with a greater number ofT2DM and NAFLD patients might be valuable in examining the predictive role of personality in metabolic characteristics. Only self-reported evaluation instruments were used. However, the questionnaires applied in this study are well-known and well-regarded research instruments. Additionally, this study’s regression models were adjusted for gender, years since diagnosis of T2DM, and current hypoglycemic treatment to reduce the effect of those variables’ bias. It should also be noted that the study period encompassed the 2019 coronavirus (COVID-19) disease’s first lockdown I outbreak on 16th March 2020. However, since a subset of the patients had already been evaluated and personal dispositions (rather than situational psychosocial features) were the primary variables of interest, the original study design was maintained. However, prior research revealed that the COVID-19 outbreak and associated lifestyle modifications may have affected T2DM patients’ quality of life [53].

Even if the number of statistically significant results in relation to all measured is relatively small, it does not mean that personality factors do not affect the NAFLD and T2DM course. It should be supposed that some part of the connections was not revealed in this analysis, in connection with multifactorial conditions of the disease and, thus, the suspected relatively weak effect of individual personality factors.

## 6. Conclusions

Our study revealed the relationship between personality factors, T2DM, and NAFLD management; a few of the correlations became significant, including links between emotional intelligence and BMI, coping styles and WHR, and fasting plasma glucose. In addition, relationships between psychological factors and lipid profile were also observed. Improving metabolic condition in the mentioned aspects may lead to the alleviation of NAFLD and T2DM states, and improve patient life. As lifestyle modifications are the main method of NAFLD treatment, and are an important factor in dealing with T2DM, willingness to change, which is affected by personality factors, is important to emphasize.

This study sheds a new light on the management of a patient with NAFLD and T2DM, pointing to the possibility of introducing psychological interventions to improve metabolic condition. However, future studies on larger groups of patients, and implementing interventions are needed to better understand the subsoil of these correlations.

## Figures and Tables

**Table 1 jcm-12-04468-t001:** Clinical and laboratory characteristics of the studied patients diagnosed with T2DM, compared between men and women.

	Women (N = 65)		Men (N = 35)			
	M	SD	M	SD	F	*p*
Age	65.39	8.82	61.77	7.94	4.091	0.046
T2DM duration [years]	7.92	5.30	9.14	11.29	0.365 *	0.549
BMI [kg/m^2^]	33.86	5.95	32.74	4.75	0.921	0.340
WHR	0.95	0.06	1.03	0.07	36.182	<0.001
FPG [mg/dL]	124.25	19	121.63	22.98	0.372	0.543
PPG [mg/dL]	146.85	23.72	146.94	26.77	<0.001	0.985
HbA1c [%]	7.01	1.27	7.05	1.33	0.027	0.870
Creatinine [mg/dL]	0.88	0.30	1.00	0.45	2.691	0.104
TC [mg/dL]	192.72	52.80	181.69	42.31	1.135	0.289
LDL [mg/dL]	109.63	48.07	98.74	38.48	1.332	0.251
HDL [mg/dL]	58.64	16.36	52.00	21.88	2.943	0.089
TG [mg/dL]	161.12	56.27	162.44	68.70	0.010	0.919
ALT [IU/L]	31.88	14.69	43.11	55.51	1.335	0.251
AST [IU/L]	27.36	8.80	30.95	10.81	1.381 *	0.247
GGTP [IU/L]	53.06	71.82	51.63	58.07	0.010	0.919
PLT [×10^3^/mL]	248.25	52.42	214.34	64.17	8.113	0.005
SBP [mmHg]	138.09	14.68	138.06	15.95	<0.001	0.991
DBP [mmHg]	79.72	12.16	85.11	10.56	4.890	0.029
Uric acid [mg/dL]	5.75	1.63	5.71	1.39	0.014	0.907
FLI	84.31	13.48	86.16	11.32	0.480	0.490
HIS	47.36	7.22	45.55	9.44	1.153	0.286
FIB-4	1.14	0.21	1.13	0.23	0.070	0.792
Current hypoglicemic drugs:	N	%	N	%	Chi^2^	*p*
Metformin	52	83%	26	74%	1.099	0.295
Sulfonylurea	25	38%	9	26%	1.647	0.199
SGLT2-inhibitor	8	12%	4	11%	0.017 **	0.999
DPP-4 inhibitor	7	11%	1	3%	1.935 **	0.255
GLP-1 analogue	2	3%	0	0%	1.099 **	0.540
Thiazolidinedione	1	2%	0	0%	0.544 **	0.999
Acarbose	2	3%	0	0%	1.099 **	0.540
Insulin	16	25%	14	40%	2.564	0.109

T2DM—type 2 diabetes mellitus, BMI—body mass index, WHR—waist-hip ratio, FPG—fasting plasma glucose, PPG—postprandial plasma glucose, HbA1c—glycated heamoglobin, TC—total cholesterol, LDL—low-density lipoprotein, HDL—high-density lipoprotein, TG—triglycerides, ALT—alanine aminotransferase, AST—asparagine aminotransferase, GGTP—gamma-glutamyltransferase, PLT—blood platelets, SBP—systolic blood pressure, DBP—diastolic blood pressure, FLI—Fatty Liver Index, HSI—Heapatic Staetosis Index, FIB-4—fibrosis-4 index, DDP-4—dipeptidyl peptidase-4, GLP-1—glucagon-like peptide-1; M—mean, SD—standard deviation, F—F test statistics (or * Welch’s test in case of non-homogeneous variance between groups), *p*—probability in the test, N—number of observations, %—percentage, Chi2—chi-squared test (or ** Fisher’s exact test in case of expected frequencies less than 5).

**Table 2 jcm-12-04468-t002:** Personality characteristics of the studied patients diagnosed with type 2 diabetes mellitus, compared between men and women.

		Women (N = 65)		Men (N = 35)			
		M	SD	M	SD	F	*p*
EPQ-R	Extraversion	10.05	3.46	9.34	3.66	0.903	0.344
	Neuroticism	13.19	5.76	10.97	5.42	3.495	0.065
	Psychoticism	6.72	2.98	6.09	2.74	1.099	0.297
	Lie	13.22	3.11	11.71	3.68	4.668	0.033
IVE	Impulsiveness	4.66	2.36	4.71	2.74	0.010	0.920
	Venturesomeness	3.79	1.94	5.11	2.51	8.672	0.004
	Empathy	12.22	2.27	10.77	3.18	6.885	0.010
INTE	General score	120.57	13.09	120.26	10.72	0.015	0.904
	Utilizing emotion	60.68	7.23	62.51	6.46	1.580	0.212
	Emotion perception	42.29	5.77	41.20	5.18	0.875	0.352
CISS	Task-oriented	56.17	7.74	56.77	6.30	0.156	0.694
	Emotion-oriented	42.77	7.82	40.51	9.35	1.646	0.202
	Avoidance	45.82	8.16	44.51	8.13	0.580	0.448
	Distraction	20.95	4.86	19.77	5.52	1.224	0.271
	Social diversion	16.46	3.35	16.51	3.37	0.006	0.940
FCZ-KT(R)	Sensory sensitivity	44.03	5.48	42.83	4.68	1.207	0.275
	Endurance	34.39	7.12	35.14	6.05	0.286	0.594
	Emotional reactivity	41.52	7.56	37.77	5.09	8.693 *	0.004
	Activity	33.25	5.73	33.54	5.70	0.061	0.805
	Briskness	40.95	5.94	40.57	4.80	0.107	0.744
	Perseveration	40.09	5.91	39.23	3.97	0.755 *	0.387
	Rhythmicity	25.57	4.42	25.83	5.13	0.070	0.792

EPQ-R—Eysenck Personality Questionnaire—Revised, IVE—Eysenck’s Impulsiveness-Venturesomeness-Empathy Questionnaire, INTE—Emotional Intelligence Questionnaire, CISS—Coping Inventory for Stressful Situations, FCZ-KT(R)—Strelau’s Formal Characteristics of Behavior—Temperament Questionnaire. M—mean, SD—standard deviation, N—number of observations, F—F test statistics (or * Welch’s test in case of non-homogeneous variance between groups), *p*—probability in the test.

**Table 3 jcm-12-04468-t003:** Results of multivariate linear regression models predicting selected metabolic parameters with personality factors. Presented as semi-partial coefficients (sizes of effect).

		BMI	WHR	FPG	PPG	HbA1C	TC	LDL	HDL	TG	FLI	HSI	FIB-4
EPQ-R	Extraversion	0.078	−0.054	0.142	0.082	0.126	0.067	−0.006	−0.179	0.067	0.040	−0.110	0.019
	Neuroticism	0.052	0.074	−0.019	−0.030	−0.074	−0.040	−0.042	0.026	0.027	0.088	0.195	−0.064
	Psychoticism	0.111	−0.027	−0.015	0.163	0.080	0.082	0.025	−0.163	0.068	0.048	−0.044	−0.081
	Lie	−0.053	0.035	−0.068	−0.004	0.001	0.060	0.100	−0.130	−0.005	−0.079	−0.171	0.030
IVE	Impulsiveness	0.050	−0.037	0.050	0.097	0.135	−0.186	−0.103	−0.085	0.084	−0.086	−0.156	0.140
	Venturesomeness	−0.029	0.041	−0.067	−0.099	−0.053	0.055	0.072	0.134	−0.108	0.026	0.049	−0.043
	Empathy	−0.108	−0.070	0.076	−0.036	−0.033	0.112	0.018	0.068	0.029	−0.076	−0.059	−0.108
INTE	Utilizing emotion	−0.250 *	0.127	0.062	0.057	0.110	0.085	0.136	0.161	−0.046	−0.150	−0.123	−0.010
	Emotion perception	0.209 *	−0.037	−0.062	−0.007	−0.040	−0.075	−0.126	−0.068	0.063	0.147	0.141	0.095
CISS	Task-oriented	0.028	−0.222 *	−0.219 *	−0.001	−0.175	−0.115	0.092	−0.036	−0.063	−0.131	0.033	0.064
	Emotion-oriented	−0.023	0.197 *	0.066	0.101	0.137	0.010	−0.043	0.039	0.023	0.096	−0.038	0.045
	Distraction	0.076	−0.196 *	−0.181	−0.098	−0.091	0.123	0.027	−0.011	−0.035	0.056	0.112	0.006
	Social diversion	0.173	−0.085	0.005	0.032	−0.112	−0.186	−0.148	−0.241 *	0.008	0.152	0.026	−0.207 *
FCZ-KT(R)	Sensory sensitivity	0.059	−0.063	−0.065	−0.136	0.013	0.031	0.086	−0.055	−0.138	0.047	−0.059	0.039
	Endurance	0.141	0.048	0.096	0.019	−0.133	−0.086	−0.149	−0.012	0.064	0.107	0.221	−0.434 *
	Emotional reactivity	0.114	0.004	0.111	0.065	0.093	0.027	0.013	−0.031	0.103	0.045	0.037	−0.061
	Activity	−0.092	0.088	0.017	0.066	0.018	−0.007	−0.001	0.083	0.004	−0.101	−0.041	0.151
	Briskness	−0.003	0.064	−0.029	0.082	0.136	0.177	−0.001	0.150	0.131	0.033	0.001	0.054
	Perseveration	0.029	−0.127	−0.083	−0.076	−0.120	−0.088	−0.132	0.026	−0.119	−0.049	−0.049	−0.084
	Rhythmicity	−0.083	0.047	0.092	−0.010	0.013	0.071	−0.131	0.143	−0.333 *	−0.141	−0.045	0.050
	R^2^	0.123	0.284	0.338	0.041	0.147	0.031	0.005	0.111	0.007	0.058	0.065	0.081
	F	1.463	2.309	1.177	1.143	1.570	0.901	1.017	1.411	0.978	1.202	1.230	1.290
	*p*	0.098	0.002	0.285	0.318	0.063	0.615	0.463	0.121	0.512	0.261	0.237	0.191

Intercept is included in each model, but not shown. Each model is adjusted for sex and years since diagnosis of diabetes, current hypoglycemic treatment (with metformin, sulfonylurea, SGLT2-inhibitors, DDP4-inhibitors, GLP-1 analogues, acarbose, insulin, thiazolidinedione). EPQ-R—Eysenck Personality Questionnaire—Revised, IVE—Eysenck’s Impulsiveness-Venturesomeness-Empathy Questionnaire, INTE—Emotional Intelligence Questionnaire, CISS—Coping Inventory for Stressful Situations, FCZ-KT(R)—Strelau’s Formal Characteristics of Behavior—Temperament Questionnaire. BMI—body mass index, WHR—waist-hip ratio, FPG—fasting plasma glucose, PPG—postprandial plasma glucose, HbA1c—glycated hemoglobin, TC—total cholesterol, LDL—low-density lipoprotein, HDL—high-density lipoprotein, TG—triglycerides, FLI—Fatty Liver Index, HSI—Hepatic steatosis index, FIB-4—fibrosis-4 index, PLT—blood platelets, R^2^—coefficient of determination, F—F test statistics (goodness of fit), *p*—probability in the test, * *p* < 0.05.

## Data Availability

The data presented in this study are available on request from the corresponding author. The data are not publicly available due to ethical reasons.

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
