# Peer review of "The Association between Personality Factors and Metabolic Parameters among Patients with Non-Alcoholic-Fatty Liver Disease and Type 2 Diabetes Mellitus—A Cross-Sectional Study"

_jcm, 2023, doi:10.3390/jcm12134468_

Round 1
Reviewer 1 Report
In the present manuscript the Authors report the results of a cross-sectional study investigating the association between psychological features and metabolic parameters among patients with T2D and NAFLD.
I have the following comments:
1. English is sub-standard and needs improvement. I suggest seeking help from a native English speaker.
2. Introduction is too long, I suggest shortening it and keep it focused on the research question.
3. The Authors list a large number of exclusion criteria. It is not clear, however, how these conditions were excluded. Was it only based on patients’ history or did you perform tests (e.g. viral hepatitis serology etc…)
4. It is unclear whether a patients without steatosis at US but with positive FLI or HSI would still be included in the present study. I suggest only evaluating patients with positive ultrasound as the use of noninvasive markers in clinical practice to diagnose steatosis is not recommended by current guidelines.
5. I suggest renaming gliflozin “SGLT2-inhibitor” and glitazon “thiazolidinediones or TZD”
6. It would be interesting to show distribution of liver fibrosis in this population (for example by calculating the FIB-4 score).
7. I suggest mentioning a recent meta-analysis showing that patients with T2D not only have a high prevalence of steatosis, but most importantly of advanced liver fibrosis (doi: 10.1016/j.diabres.2022.109981)
Needs quite a bit of editing.
Author Response
Response to Reviewer 1 Comments
Thank the Reviewer’s very much for your time and valuable comments on our manuscript. The responses for all points are below. The changes were introduced into the text of manuscript, as suggested by the Reviewer.
Point 1: English is sub-standard and needs improvement. I suggest seeking help from a native English speaker.
Response 1: Thank you very much for this attention. Our article has been checked and corrected according to the recommendations of a antive English speaker
Point 2: Introduction is too long, I suggest shortening it and keep it focused on the research question.
Response 2: Thank you very much for this attention. In agreement, we made the changes in the manuscript.
Point 3: The Authors list a large number of exclusion criteria. It is not clear, however, how these conditions were excluded. Was it only based on patients’ history or did you perform tests (e.g. viral hepatitis serology etc…)
Response 3: Thank you very much for this attention. All potential secondary causes of fatty liver were excluded in each patient. Medical history was taken into account, including medication use, alcohol consumption, and signs and symptoms suggestive of conditions that may be related to fatty liver. In case of diagnostic doubts, we performed additional tests - e.g. we determined the level of growth hormone or thyroid hormones. We performed virological tests in each patient to exclude infection with hepatotropic viruses.
Point 4: It is unclear whether a patients without steatosis at US but with positive FLI or HSI would still be included in the present study. I suggest only evaluating patients with positive ultrasound as the use of noninvasive markers in clinical practice to diagnose steatosis is not recommended by current guidelines.
Response 4: In accordance with this remark. Patients who had both features of fatty liver in the US examination and HSI value ≥ 36, FLI ≥ 60 and FIB-4 < 1.3 were qualified for the study. In our opinion, the introduction of such a diagnostic panel for NAFLD increases the sensitivity and specificity of diagnosing this pathology.
Point 5: I suggest renaming gliflozin “SGLT2-inhibitor” and glitazon “thiazolidinediones or TZD”
Response 5: Thank you very much for this attention. In agreement, we made the changes in the manuscript.
Point 6: It would be interesting to show distribution of liver fibrosis in this population (for example by calculating the FIB-4 score).
Response 6: Thank you for this remakr, we agree that the article would benefit from additional characteristics regarding liver fibrosis in the group. Descriptive statistics of FIB-4 among men and women was added to Table 1. Also, a regression model predicting the FIB-4 score was calculated and summarized in Table 3 with the remaining models. An increase in the FIB-4 score was linked to a fall in endurance and social diversion.
Point 7: I suggest mentioning a recent meta-analysis showing that patients with T2D not only have a high prevalence of steatosis, but most importantly of advanced liver fibrosis (doi: 10.1016/j.diabres.2022.109981)
Response 7: Thank you very much for this attention. In agreement, we made the changes in the manuscript.
In addition, we modified our article according to the suggestions of other reviewers.
We sincerely hope that all changes introduced by us in the text will be fully satisfactory for the Reviewer.

Reviewer 2 Report
dear authors,
thank you for submitting this paper to JCM.
The topic of psychiatric comorbidities in NAFLD is timely and interesting. However, current paper needs many revisions.
Introduction is by far too long and disorganised. This will need to be cut down in words and to read smoothly and focusing on the topic of the paper.
The authors make conclusions based on simple differences between groups. This is not enough to verify a statistical association. Please, move on to multivariate analysis and correct for confounding factors, to make the statistical analysis sound and strong.
nothing to suggest
Author Response
Response to Reviewer 2 Comments
Thank the Reviewer’s very much for your time and valuable comments on our manuscript. The responses for all points are below. The changes were introduced into the text of manuscript, as suggested by the Reviewer.
Point 1: Introduction is by far too long and disorganised. This will need to be cut down in words and to read smoothly and focusing on the topic of the paper.
Response 1: Thank you very much for this attention. In agreement, we made the changes in the manuscript.
Point 2: The authors make conclusions based on simple differences between groups. This is not enough to verify a statistical association. Please, move on to multivariate analysis and correct for confounding factors, to make the statistical analysis sound and strong.
Response 2: We agree that the statistical analysis starts with integroup comparisons - between men and women in the studied group. However, the main part of the analysis is found in table 3 and it comprises results of a series of multivariate linear regression models, which predict selected metabolic variables based on selected personality factors. The table shows semi-partial corelations (i.e. measures of effect sizez), with statistically significant associations marked with an asterisk (*). We put a greater emphasis on the multivariate character of the constructed models. We hope that the table will be most informative in this form.
In addition, we modified our article according to the suggestions of other reviewers.
We sincerely hope that all changes introduced by us in the text will be fully satisfactory for the Reviewer.

Round 2
Reviewer 1 Report
I find the manuscript improved. No further comments.
I believe the Authors adequately addressed the points I made. English can still be improved.
Reviewer 2 Report
no further comments
no further comments